# Underwater Acoustic Target Recognition: A Combination of Multi-Dimensional Fusion Features and Modified Deep Neural Network

**Xingmei Wang [1],\*, Anhua Liu [1], Yu Zhang [2] and Fuzhao Xue [1]**

[1]   College of Computer Science and Technology, Harbin Engineering University, Harbin 150001, China
[2]   College of Computer Science and Technology, Harbin Institute of Technology, Harbin 518000, China
\*   Correspondence: wangxingmei@hrbeu.edu.cn; Tel.: +86-189-4505-5955

**Abstract:** A method with a combination of multi-dimensional fusion features and a modified deep neural network (MFF-MDNN) is proposed to recognize underwater acoustic targets in this paper. Specifically, due to the complex and changeable underwater environment, it is difficult to describe underwater acoustic signals with a single feature. The Gammatone frequency cepstral coefficient (GFCC) and modified empirical mode decomposition (MEMD) are developed to extract multi-dimensional features in this paper. Moreover, to ensure the same time dimension, a dimension reduction method is proposed to obtain multi-dimensional fusion features in the original underwater acoustic signals. Then, to reduce redundant features and further improve recognition accuracy, the Gaussian mixture model (GMM) is used to modify the structure of a deep neural network (DNN). Finally, the proposed underwater acoustic target recognition method can obtain an accuracy of 94.3% under a maximum of 800 iterations when the dataset has underwater background noise with weak targets. Compared with other methods, the recognition results demonstrate that the proposed method has higher accuracy and strong adaptability.

**Keywords:** multi-dimensional fusion features; gaussian mixture model; deep neural network; underwater acoustic target recognition

## 1. Introduction

With the development of sonar technology, underwater acoustic target recognition has become one of the major functions of sonar systems. It is extensively used for marine biological survey, marine exploration and other scientific activities. However, existing underwater acoustic target recognition still depends on the decisions of well-trained solar persons. It has been difficult to implement continuous monitoring and recognition [1,2]. Therefore, underwater acoustic target recognition with a high recognition accuracy and efficiency attracts extensive attention both in military and civil fields [3–8].

Currently, underwater acoustic target recognition contains feature extraction and recognition. Feature extraction is a process which extracts various features from underwater acoustic signals [9,10]. The Mel filter bank has been widely used in feature extraction. It was designed for imitating the band pass filter bank features of the human ear, and it has been the foundation of most speech processing, such as underwater acoustic target recognition, speaker recognition [11–15]. The cepstral or energy features are obtained from the Mel filter bank, which is known as the Mel frequency cepstral coefficient (MFCC). It has been considered the baseline feature for most applications. Lim, T et al. firstly introduced MFCC to recognize underwater acoustic targets. Experimental results demonstrate that the method is very promising for underwater acoustic target recognition [11]. Later Lim and Bae et al. proposed a method which combined the MFCC with neural network. This method also

achieved a good result on the experimental dataset [12]. However, the MFCC cannot describe the optimal features for all underwater acoustic targets [16,17]. To solve the disadvantages, the Gammatone frequency cepstral coefficient (GFCC) was introduced into underwater acoustic target recognition [18]. The GFCC based on the Gammatone filter bank can better extract voiceprint features of the underwater acoustic target [18,19]. In addition, feature parameters of the GFCC can improve the recognition rate and robustness under a noisy background [17–21]. In 2014, Zeng and Wang et al. applied GFCC and the Hilbert–Huang transform (HHT) to recognize underwater acoustic targets, which achieved better results than the MFCC. Since then, the GFCC has gradually entered the field of research [22–27]. Although the recognition results have been modified, it cannot accurately describe original underwater acoustic signals by only extracting one type of feature. In 2018, Sharma et al. proposed a new method of combining the MFCC and modified empirical mode decomposition (MEMD) to describe mixed multi-dimensional features of original acoustic signals [16]. In feature extraction, MEMD has been mapped to Hilbert space through HHT, the instantaneous energy (IE) and instantaneous frequency (IF) of the intrinsic mode function (IMF) were extracted, and good results have been achieved through the combination with the MFCC [16,21–24,28–31]. Inspired by these, a multi-dimensional fusion features method is proposed in this paper. The method combines the advantages of the GFCC with MEMD, and the original underwater acoustic signals can be described from multiple angles.

Recognition is a process of identifying an unknown target signal, which trains the observed target signals according to a certain method. The Gaussian mixture model (GMM) is widely used in the field of recognition and has achieved good results. In 2011, Kotari et al. proposed a method to recognize underwater mines, which is based on the GMM and achieved a good result [32]. On this basis, there are various methods, which have been proposed to improve GMM in underwater acoustic target recognition. Recently, Wang et al. proposed a novel method that systematically combines the deep Boltzmann machine (DBM) with the Dirichlet process based Gaussian mixture model (DP-GMM) to bypass the problem of distribution mismatch. The method can improve the effectiveness and the robustness of the GMM [33,34]. However, compared with the deep learning method, the GMM recognition effect is unsatisfactory in the field of multi-dimensional features [9,10]. GMM is a shallow recognition model, which cannot extract the deep information features of acoustic signals and lacks the ability to obtain the essential features of a large dataset. However, deep neural network (DNN) is a deep recognition model, which can extract abstract and invariant features from a large dataset. Recently, the DNN has gradually attracted increased attention in underwater acoustic target recognition. In 2018, Ibrahim, Ali k et al. recognized the grouper species by their sounds using DNN, and the results are significantly better than the other previously reported methods [35]. Compared with the machine learning model, it can bring great improvement in recognition accuracy. Moreover, the DNN has a strong robustness to recognize underwater acoustic signals under noise conditions. Recognition results have further demonstrated that the DNN has a better learning capability than the GMM [36,37]. However, when the DNN directly recognizes features, it can generate many redundant features which greatly wastes computing resources. In this paper, to overcome these shortcomings, a modified DNN is proposed to accurately recognize underwater acoustic targets. The GMM is used to extract the statistical parameters of the feature matrix, which modify the structure of the DNN.

This paper proposes an MFF-MDNN method which combines multi-dimensional fusion features with a modified DNN to recognize underwater acoustic targets. For solving the problem whereby a single feature cannot better describe underwater acoustic signals, a multi-dimensional fusion features method is given in this paper, which uses the GFCC and MEMD to extract multi-dimensional features. Then, a dimension reduction method is proposed to ensure the same time dimension. It can obtain multi-dimensional fusion features in the original underwater acoustic signals. In addition, the problem of many redundant features being generated in the processing recognition in the DNN was solved by a method utilizing a modified DNN in this paper. The GMM is used to modify the structure of the DNN to improve the accuracy of recognition. The proposed underwater acoustic target recognition method was applied on a dataset, which was divided into six categories, including four types of ships,

underwater mammals, and underwater background noise with weak targets. The recognition results show that the method has good effectiveness and adaptability. The proposed method has important theoretical and practical value.

The rest of the paper is organized as follows: Section 2 discusses the multi-dimensional fusion features method. Section 3 describes a modified deep neural network. Section 4 presents the experimental results and analysis. Finally, Section 5 summarizes and concludes this work.

## 2. Multi-Dimensional Fusion Features Method

### 2.1. Gammatone Frequency Cepstral Coefficient

The GFCC is one of mainstream feature extraction algorithms in recognition [18,19], Figure 1 shows the specific GFCC process. To obtain the power spectrum, the discrete Fourier transform (DFT) is employed to transform the original signals from the time domain to the frequency domain. The frequency is distorted from the Hertz scale to the Gammatone scale using the Gammatone filter bank. The Gammatone power spectrum is taken as a logarithm. Finally, the GFCC features are extracted by discrete cosine transforms (DCT).

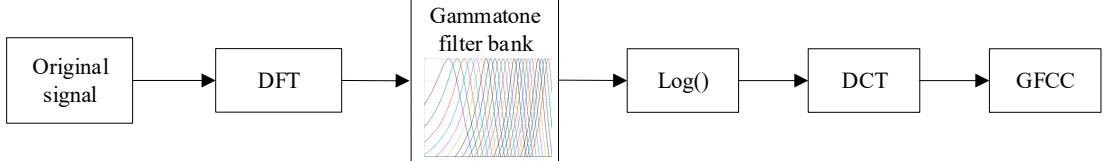

**Figure 1.** The process of the Gammatone frequency cepstral coefficient (GFCC).

The DFT is obtained by the Fourier transformation as described by the following equation:

$$x(k) = \sum_{t=0}^{M_S-1} x(t)e^{-j2\pi tk/M_S}, \quad 0 \le k \le M_S, \tag{1}$$

where $x(t)$ is the original underwater acoustic signal, $M_S$ is the needed number of the DFT.

The Gammatone filter refers to a causal filter with an infinitely long sequence impulse response. In the filter bank, the time domain impulse response of each Gammatone filter can be considered as the product of the Gammatone function, and the acoustic signals can be calculated as below:

$$g_i(k) = k^{n-1} \exp(-2\pi B_i k) \cos(2\pi f_i + \varphi_i)u(k), \ 1 \le i \le N, \tag{2}$$

where $n$ is the order of the filter, $B_i$ is the attenuation factor of the filter, $f_i$ is the center frequency, $\varphi_i$ is the phase of the filter, $u(k)$ is the step function, $N$ is the total number of filters.

When extracting the features from the underwater acoustic signals, the bandwidth of each filter is determined by the critical band of the human ear for simulating human auditory features. The critical frequency band is expressed as

$$ERB(f_i) = 24.7 \times (4.37 f_i/1000 + 1), \tag{3}$$

$$b_i = 1.019 ERB(f_i), \tag{4}$$

where $b_i$ is the bandwidth of each sub-band filter in the Gammatone filter bank, which is obtained from the critical band.

The energy spectrum $E_S(i)$ is shown as

$$E_S(i) = \ln\left[\sum_{k=0}^{Q-1} |x(k)|^2 g_i(k)\right], \tag{5}$$

where $Q$ is the number of filters and $i = 1, 2, \ldots, Q, \{k_{b_i}\}_{i=0}^{Q+1}$ is the boundary of the filters.

The DCT is applied to calculate the logarithm of the filter bank as described by the following equation:

$$GFCC(n) = \sum_{k=0}^{Q-1} E_S(i) \cos\left(\frac{(\pi n(i - 0.5))}{Q}\right), \qquad 0 \le n \le Q - 1. \tag{6}$$

To verify the validity of the proposed multi-dimensional fusion features method in this paper, Figure 2 shows the time-domain waveform of the original underwater acoustic signals, which is a section of underwater mammals in the dataset. The feature extraction results of original underwater acoustic signals are given in Figure 3 on the original underwater acoustic signals shown in Figure 2.

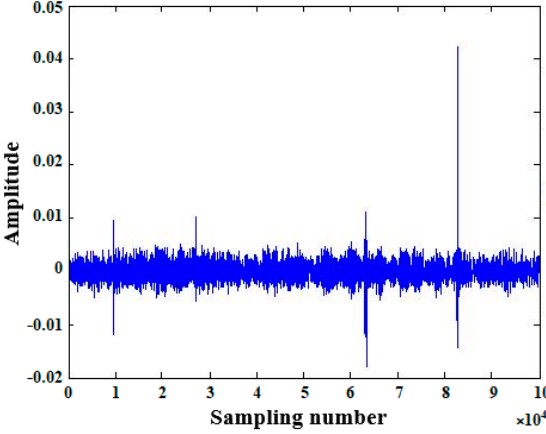

**Figure 2.** Original underwater acoustic signals.

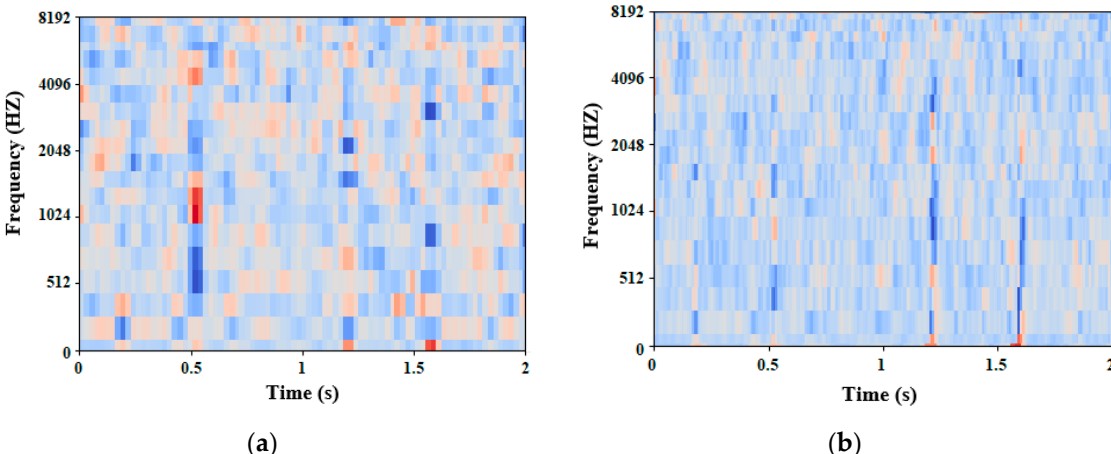

(a)                (b)

**Figure 3.** Features extraction results of the original underwater acoustic signals: (**a**) Features of the Mel frequency cepstral coefficient (MFCC) in original underwater acoustic signals; (**b**) Features of the GFCC in original underwater acoustic signals.

As depicted in Figure 3, because the MFCC pays more attention to semantic features, feature extraction results of original underwater acoustic signals are rough in Figure 3a. The GFCC extract

features of original underwater acoustic signals in Figure 3b, which can remove considerable redundant noise information in underwater acoustic signals, improve robustness, and retain effective voiceprint feature. Moreover, the extracted features are more precise and accurate.

To further compare the anti-noise performance of GFCC and MFCC in the feature extraction process, Gaussian white noise was added into Figure 2. When the signal-to-noise ratio was 10 dB, the time domain waveform of noise signals is shown in Figure 4. Figure 5 shows the feature extraction results of the MFCC and GFCC algorithm on the original underwater acoustic signals shown in Figure 4.

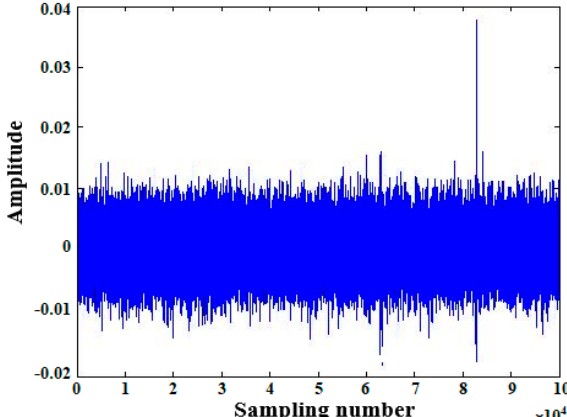

**Figure 4.** Underwater acoustic signals with a signals-to-noise ratio of 10 dB.

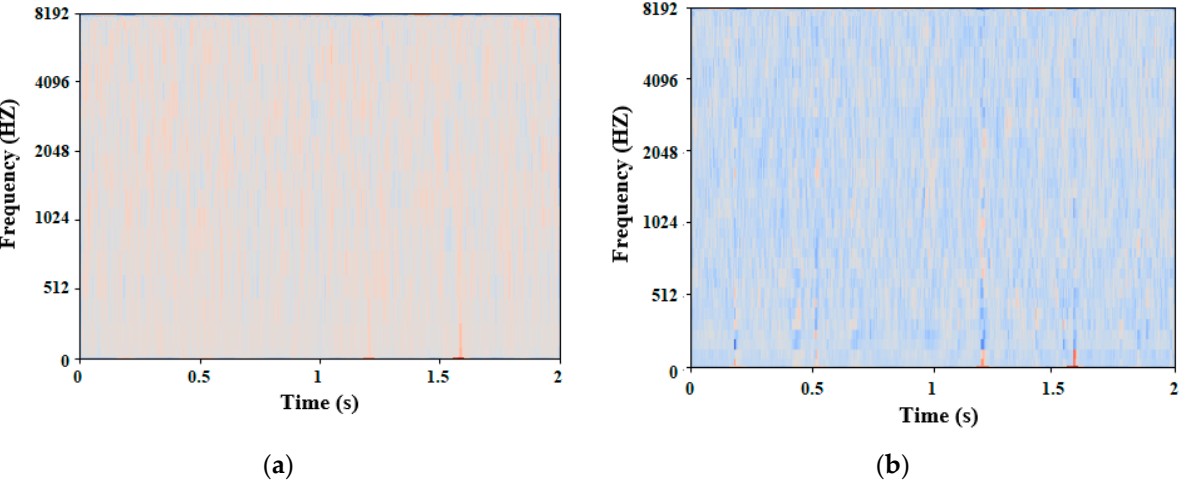

(**a**)                                          (**b**)

**Figure 5.** Features extraction results of underwater acoustic signals with a signals-to-noise ratio 10 dB: (**a**) Features of MFCC in underwater acoustic signals with a signals-to-noise ratio 10 dB; (**b**) Features of the GFCC in underwater acoustic signals with a signals-to-noise ratio 10 dB.

It can be seen from Figure 5a,b that the GFCC algorithm has strong anti-noise performance and has superiority in the feature extraction of underwater acoustic signals targets. It is beneficial to subsequent recognition research. But the features, distributions, and sizes extracted by the MFCC change significantly and it affects the accuracy of underwater acoustic target recognition.

Meanwhile, to verify the effectiveness and adaptability of the GFCC again, Table 1 shows MFCC-GMM and GFCC-GMM recognition accuracy when dataset did not have underwater background noise with weak targets.

**Table 1.** The estimated optimal threshold and $\eta$ of 20 different underwater sonar images.

| Experiment Times | MFCC-GMM | GFCC-GMM |
|---|---|---|
| 1 | 0.85 | 0.94 |
| 2 | 0.88 | 0.90 |
| 3 | 0.88 | 0.90 |
| 4 | 0.86 | 0.93 |
| 5 | 0.87 | 0.92 |
| 6 | 0.89 | 0.92 |
| 7 | 0.89 | 0.95 |
| 8 | 0.88 | 0.92 |
| 9 | 0.87 | 0.92 |
| 10 | 0.90 | 0.90 |

It can be seen from Table 1, the recognition accuracy of the GFCC-GMM is higher than that of MFCC-GMM. The GFCC can better describe underwater acoustic signals. Therefore, The GFCC has better robustness than the MFCC, and extracting voiceprint features has better adaptability to underwater acoustic target recognition.

### 2.2. Modified Empirical Mode Decomposition

MEMD is an improved algorithm for empirical mode decomposition (EMD). It refers to a signal decomposition method based on the local features of signals. The method exploits the advantages of the wavelet transform. Meanwhile, it solves the problem of selecting a wavelet basis and determining the decomposition scale in the wavelet transform [25]. Therefore, it is more suitable for non-linear and non-stationary signals analysis. MEMD is a white-adapted signal decomposition method, which can be employed for the analysis of underwater acoustic signals. According to empirical mode decomposition, it is assumed that all complex signals consist of simple IMF, and each IMF is independent of each other. The algorithm process of MEMD is shown in Figure 6. It can decompose the different scales or trend components of underwater acoustic signals step by step and produce a series of data sequences with the same feature scales. Compared with the original underwater acoustic signals, the decomposed sequence has stronger regularity.

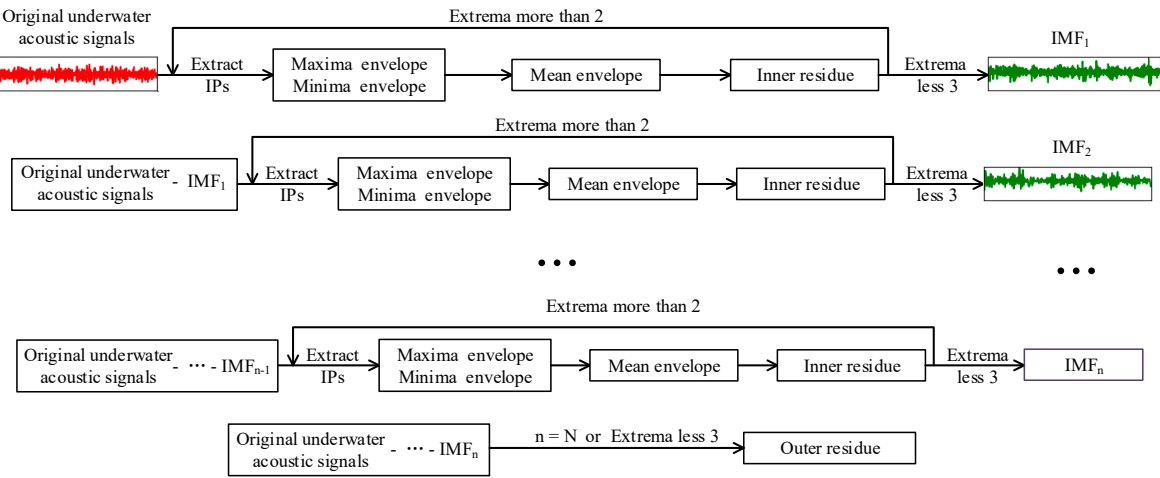

**Figure 6.** The flowchart of modified empirical mode decomposition (MEMD).

MEMD calculates the x-coordinates of the interpolation points (IPs) by the following equation:

$$z(t) = \frac{d^3}{dt^3}x(t), \quad t_{\max} = \left\{t : \frac{d}{dt}z(t) = 0, \frac{d^2}{dt^2}z(t) < 0\right\}$$
$$t_{\min} = \left\{t : \frac{d}{dt}z(t) = 0, \frac{d^2}{dt^2}z(t) > 0\right\} \tag{7}$$

where $t_{max}$ and $t_{min}$ contain the IPs.

To extract the IE and IF, IMFs are usually transformed by the HHT as below:

$$H(t) = \frac{1}{\pi} \int_{-\infty}^{+\infty} \frac{\overset{\wedge}{H}(\tau)}{t - \tau} d\tau, \tag{8}$$

where $\overset{\wedge}{H}(\tau)$ is the IMF.

The analytic signals can be calculated as follows:

$$V(t) = \overset{\wedge}{H}(t) + jH(t) = a(t)e^{j\vartheta(t)}, \tag{9}$$

where $a(t)$ is the model of $H(t)$, $j$ is the imaginary unit.

$a(t)$ is described by the following equation:

$$a(t) = \sqrt{H(t)^2 + \overset{\wedge}{H}(t)^2}, \tag{10}$$

where $\vartheta(t)$ is the phase of $H(t)$, it can be calculated as below:

$$\vartheta(t) = \arctan\left(\frac{\overset{\wedge}{H}(t)}{H(t)}\right). \tag{11}$$

The IE $a^2(t)$ and IF can be extracted by analyzing the signals. The IF is obtained by the following equation:

$$f(t) = \frac{1}{2\pi} \frac{d\vartheta(t)}{dt}. \tag{12}$$

### 2.3. Multi-Dimensional Fusion Features Algorithm

The underwater environment is complex and changeable, so it is difficult to describe underwater acoustic signals with a single feature. GFCC and MFCC are both algorithms for simulating human hearing. The abilities to extract the infrasound or ultrasound features are limited, but the MEMD algorithm can effectively extract these features. Therefore, GFCC and MEMD were developed to extract multi-dimensional features in this paper. On this basis, to ensure the same time dimension, a dimension reduction method is proposed to obtain multi-dimensional fusion features in the original underwater acoustic signals.

### 2.3.1. Multi-Dimensional Feature Extraction

Figure 7 shows the feature extraction process. The feature extraction algorithm can extract the IF and IE from the IMFs, Meanwhile, it can extract the GFCC features and combine it with IF and IE to improve the capabilities of the algorithm. The process can eventually extract features that contain three dimensions and construct a feature vector.

The global feature vector $\vec{g}$ can be expressed as

$$\vec{g_i} = \left\{ G_i(t), a_i^2(t), f_i(t) \right\}, \tag{13}$$

where $t$ is time series (sampling points), $G_i(t)$ is the feature value of the GFCC at time $i$, $a_i^2(t)$ is IE at time $i$, $f_i(t)$ is IF at time $i$.

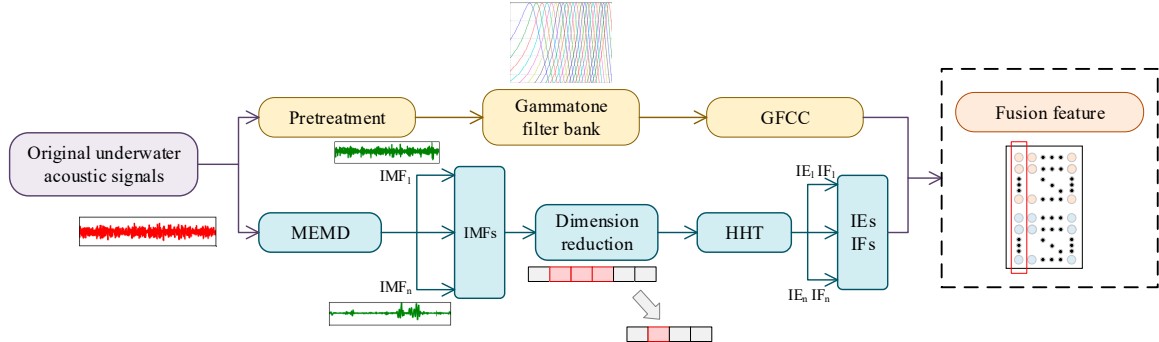

**Figure 7.** The process of feature extraction.

Figure 8 shows the structure of the feature fusion matrix. The feature fusion matrix is sorted in the same time sequence, which holds the same sequence. Furthermore, it can merge multiple features. It is noteworthy that the weight of GFCC is adjusted to the same proportion as that of IE and IF, and the ratio can be considered as a research direction in future studies. Since the features are exceedingly redundant, and the same time series are ensured, it is necessary to perform dimensionality reduction operations, which is similar to random pooling and framing for IE and IF.

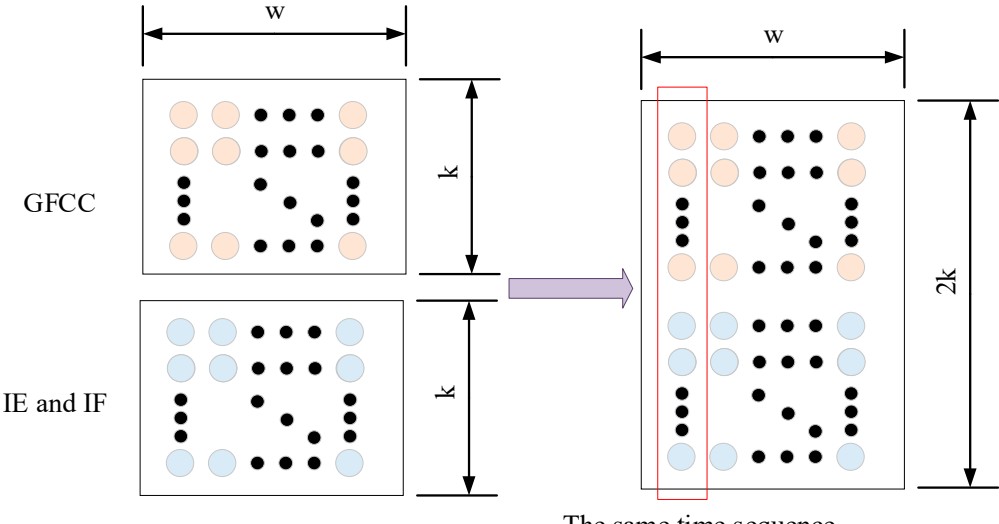

**Figure 8.** Fusion feature matrix structure.

### 2.3.2. Dimension Reduction Method

The HHT is used to extract IE and IF on IMFs in a certain time series (sampling points), and the GFCC divides frames before Gammatone filtering. As a result, the temporal features of IE and IF are different from those of the GFCC. Therefore, it is necessary to reduce the dimension of IE and IF.

Figure 9 shows the dimensionality reduction algorithm structure diagram. It is a one-dimensional feature (IE or IF), and the red box represents the pooled domain. The size and step size of the pooled domain is identical to the frame length when the GFCC frames are divided. The frame length equals the ratio of the one-dimensional feature (IE or IF) to the GFCC feature length.

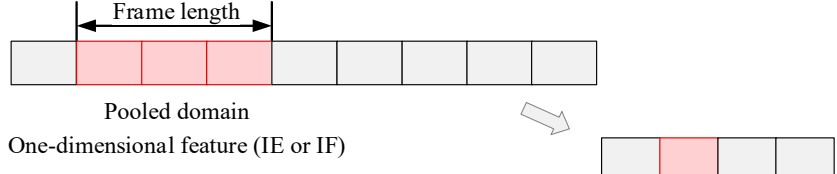

**Figure 9.** Dimensionality reduction algorithm structure diagram.

The specific process of the proposed multi-dimensional fusion features is shown as Algorithm 1.

---

**Algorithm 1** Multi-dimensional Fusion Feature

---

Input: original underwater acoustic signals $s(t)$;
Output: fusion feature vector;

Initialization: $M_S$ (the number points in the DFT), $Q$ (the number of filters), $i = 1, 2, \ldots, Q, \{k_{b_i}\}_{i=0}^{Q+1}, N, M$;

Procedure:
1: Let $x(t) \leftarrow s(t)$;
2: Calculate $x(k)$ using Equation (1);
3: Calculate $Trg_i(k)$ using Equation (2);
4: Calculate the energy spectrum $E_S(l)$ using Equation (3);
5: Calculate the $k_{b_i}$ using Equation (6);
6: Calculate the GFCC using Equation (7);
7: Let $r_0(t) \leftarrow s(t), h_k^0(t) \leftarrow r_{k-1}(t)$;
8: For $m \leftarrow 1$ to $M$ or he number of extrema in $r_k(t)$ is 2 or less do:
9:      For $n \leftarrow 1$ to $N$ do:
10:            Calculate the x-coordinates of the IPs, i.e., obtain $t_{\max}$ and $t_{\min}$ using Equation (11);
11:            Calculate the y-coordinates of the IPs, $h_k^{n-1}(t)$;
12:            $y_{\max} \leftarrow h_k^{n-1}(t_{\max}), y_{\min} \leftarrow h_k^{n-1}(t_{\min})$;
13:            Create the maxima envelope, $e_{\max}(t)$ using cubic spline interpolation, with the IPs as $\{t_{\max}, y_{\max}\}$;
14:            Create the minima envelope, $e_{\min}(t)$, using cubic spline interpolation, with the IPs as $\{t_{\min}, y_{\min}\}$;
15:            Deduce the mean envelope, $e(t) \leftarrow \frac{e_{\max}(t) + e_{\min}(t)}{2}$;
16:            $h_k^n(t) \leftarrow h_k^{n-1}(t) - e(t)$;
17:      $h_k(t) \leftarrow h_k^N(t), r_k(t) \leftarrow r_{k-1}(t) - h_k(t)$;
18: Decomposition results: $s(t) \leftarrow r_M(t) + \sum_{k=1}^{M} h_k(t)$ ($h_k(t)$ is the kth order of IMFs);
19: Dimension Reduction to IMFs;
20: Calculate the $H(t)$ using Equation (8);
21: Calculate the IE and IF using Equation (10) to Equation (12);
22: Construct fusion feature vector using Equation (13).

---

## 3. Modified Deep Neural Network

When a DNN is directly used to recognize underwater acoustic targets, due to the direct input of the features of the underwater acoustic signals, there exist many redundant features in the processing of recognition. The GMM can extract the statistical parameters of the underwater acoustic signals, which can reduce the length of the underwater acoustic signals. The GMM can reduce the size of the DNN model, can reduce redundant features, and further improve recognition accuracy. Therefore, The GMM is used to modify the structure of the DNN in this paper. The modified DNN, which is shown in Figure 10, is a fully connected feedforward neural network with multiple hidden layers, and it can better accomplish underwater acoustic target recognition tasks.

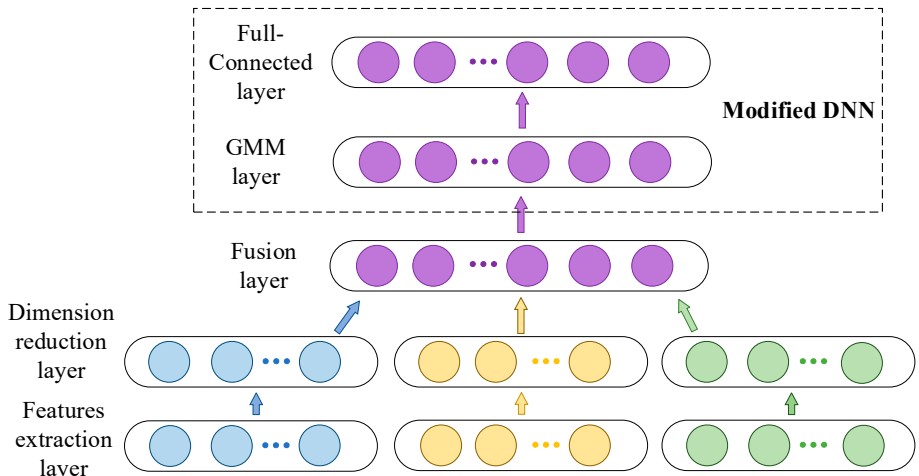

**Figure 10.** Schematic diagram of the modified deep neural network (DNN).

The Gaussian probability density function of each component is weighted to yield the GMM probability density function with order, which is expressed as

$$P(x|\lambda) = \sum_{i=1}^{M} w_i N_i(x),\tag{14}$$

where $x$ is the $D$ dimension of the eigenvector, $N_i(x), i = 1, 2, \cdots, M$ is sub-distribution, $w_i, i = 1, 2, \cdots, M$ is mixed weight.

Each sub-distribution $N_i(x)$ refers to a $D$ dimensional joint Gaussian probability distribution.

$$N_i(x) = \frac{1}{(2\pi)^{D/2}|\Sigma_i|^{1/2}} \exp\left\{-\frac{1}{2}(x-\mu_i)^{\mathrm{T}}\Sigma_i^{-1}(x-\mu_i)\right\},\tag{15}$$

where $\mu_i$ is mean vector, $\Sigma_i$ is covariance matrix, $T$ is the number of feature vectors, mixed weights $w_i, i = 1, 2, \cdots, M$ satisfies $\sum_{i=1}^{M} w_i = 1$.

The full parameters of the GMM model for each kind of underwater acoustic signal targets are composed by the mean vector of each component, covariance matrix, and the set of mixed weights. The eigenvectors corresponding to the parameters are expressed as follows:

$$\lambda = \{w_i, \mu_i, \Sigma_i\}, i = 1, 2, \cdots, M.\tag{16}$$

The feature vector sequence extracted from training data is $X = \{x_t\}, t = 1, 2, \cdots, T$ and the likelihood probability defined as

$$P(X|\lambda) = \prod_{t=1}^{T} P(x_t|\lambda).\tag{17}$$

Substituting it into the Gauss density function, as follows:

$$P(X|\lambda) = \sum_{t=1}^{T} \lg\left\{\sum_{i=1}^{M} w_i N_i(x_t, \mu_i, \Sigma_i)\right\}.\tag{18}$$

The parameter estimation method adopted in the GMM model is maximum likelihood estimation (MLE). According to the feature vector sequence extracted from training data, the parameters of the

model are adjusted continuously until the maximum likelihood probability is $P(X|\lambda)$. The maximum likelihood probability is $\lambda_i$, the model parameters are

$$\lambda_i = \arg \max P(X|\lambda). \tag{19}$$

According to Equation (16), the input parameters of the DNN are defined as

$$r = \Sigma_i w_i \times \frac{\mu_i - \mu_{i,ubm}}{(diag\{\Sigma_i\})^{1/2}}, \tag{20}$$

where $\mu_{i,ubm}$ is the mean vector of the universal background model (UBM).

Given a dimension input eigenvector, the activation vector of the first hidden layer is expressed as

$$\overrightarrow{h}^{(1)} = \sigma\left(W^{(1)\mathrm{T}}\overrightarrow{x} + \overrightarrow{b}^{(1)}\right), \tag{21}$$

where $W^{(1)\mathrm{T}}$ is the first hidden layer weight matrix transpose, dimension is $I \times N_1$, $\overrightarrow{b}^{(1)}$ is the size of the offset vector, $\sigma$ is activation function of the hidden layer.

The activation vectors of the $i$ hidden layer are obtained by the activation vectors $\overrightarrow{h}^{(i)}$ of the second hidden layer.

$$\overrightarrow{h}^{(i)}\left(\overrightarrow{x}\right) = \sigma\left(W^{(i)\mathrm{T}}\overrightarrow{h}^{(i-1)}\left(\overrightarrow{x}\right) + \overrightarrow{b}^{(i-1)}\right), \tag{22}$$

where $N_i$ is the number of neurons in the $i$ hidden layer, $W^{(i)\mathrm{T}}$ is the transpose of the weight matrix of the $i$ hidden layer, the dimension is $N_{i-1} \times N_i$, and $\overrightarrow{b}^{(i)}$ is the offset vector of size.

Since the ReLU function performs well in DNN classification and recognition tasks, the ReLU function serves as hidden layer activation functions. It is specifically defined as:

$$\sigma(a) = \begin{cases} a & if\ a > 0 \\ 0 & if\ a \le 0 \end{cases}, \tag{23}$$

The output layer of the DNN uses the SoftMax function to complete the output category. It finally realizes underwater acoustic signals classification and recognition. The SoftMax function is expressed as

$$s(z_k) = \frac{e^{z_k}}{\sum_{j=1}^{J} e^{z_j}}, \tag{24}$$

where $z_k$ is the vector of the dimension output layer.

SoftMax regression algorithm serves as the loss function.

$$J(w) = -\frac{1}{m}\left[\sum_{i=1}^{m}\sum_{j=1}^{d} 1\{y^{(i)} = d\}\log\frac{e^{w_j^T z^{(i)}}}{\sum_{l=1}^{d} e^{w_l^T z^{(j)}}}\right], \tag{25}$$

where $\log \dfrac{e^{w_j^T z^{(i)}}}{\sum_{l=1}^{d} e^{w_l^T z^{(j)}}}$ is the Logarithmic values for softmax function, $1\{\cdot\}$ is the characteristic function. $y^{(i)}$ is the predicted label value of the $i$ data in the dataset, $d$ is the label value, when $y^{(i)} = d$ is true, return 1, otherwise return 0.

## 4. Experiment Results and Analysis

This section shows numerical examples to validate the generality and effectiveness of the proposed MFF-MDNN for underwater acoustic target recognition. The dataset was divided into six categories, including four types of ships, underwater mammals, and underwater background noise with weak targets in this paper. The total length of our dataset is almost 20 h. Each acoustic signal was divided into two seconds, and the background noise dataset was used to simulate the general situation of the underwater. The train set was three times as large as the test set in the experiments.

To demonstrate the MEMD suitable for feature extraction in underwater acoustic targets, Figure 11 shows the first 5 IMFs waveforms extracted of the EMD and the MEMD on the original underwater acoustic signals shown in the section of Figure 2. Figure 11a decomposes the IMFs waveform graph of EMD. Figure 11b decomposes the IMFs waveform of the MEMD.

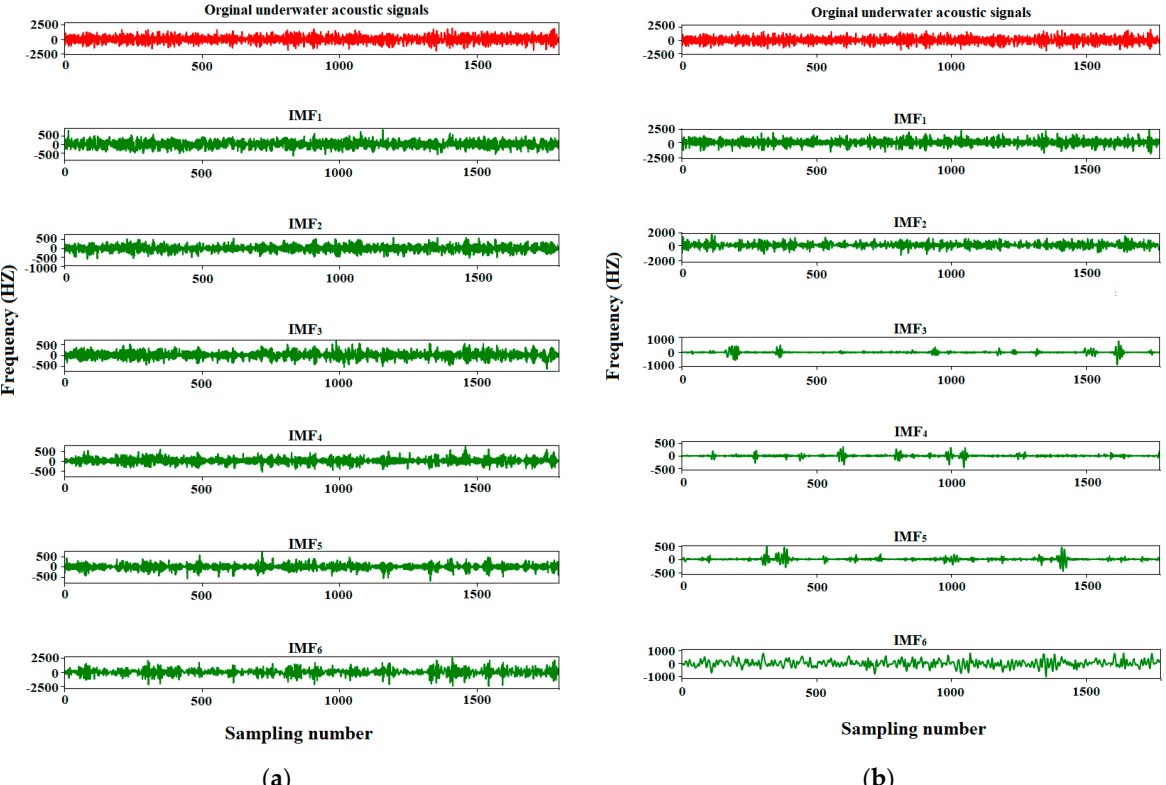

(**a**)　　　　　　　　　　　　　　　　　　　　(**b**)

**Figure 11.** The first 5 intrinsic mode functions (IMFs) waveforms extracted from the empirical mode decomposition (EMD) and MEMD: (**a**) The first 5 IMFs waveforms extracted from the EMD; (**b**) The first 5 IMFs waveforms extracted from the MEMD.

From Figure 11, it can be observed that the underwater environment was extremely complex and that the EMD shows little difference between each order IMF and the original underwater acoustic signals. While MEMD can decompose the original underwater acoustic signals more distinctly, it can improve the effectiveness of feature information extraction in underwater acoustic signals, which is conducive to the subsequent recognition of underwater targets.

Therefore, the proposed multi-dimensional fusion features method has validity to some extent from Figures 3 and 5, Table 1, and Figure 11 in this paper.

Similarly, to verify the superiority of the proposed modified DNN in this paper, Figure 12 shows the recognition accuracy of 30 experiments with a different training set and testing set in the dataset when the maximum iteration was 800, which included the GMM recognition method using the MFCC feature extraction (MFCC-GMM) [11,29], the proposed modified DNN recognition method using

MFCC feature extraction (MFCC-MDNN) [11], the proposed modified DNN recognition method using GFCC feature extraction (GFCC-MDNN) [18].

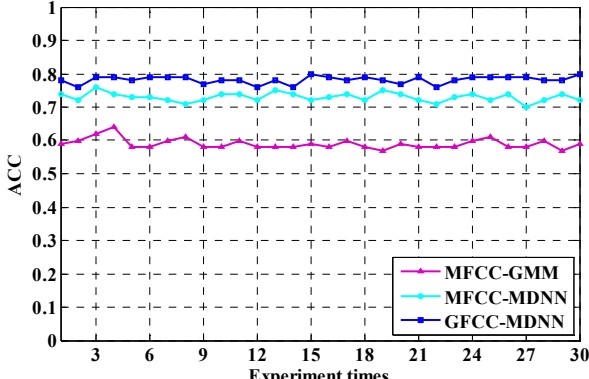

**Figure 12.** The recognition accuracy of the Mel frequency cepstral coefficient-Gaussian mixture model (MFCC-GMM), the Mel frequency cepstral coefficient-modified deep neural network (MFCC-MDNN) and the Gammatone frequency cepstral coefficient-modified deep neural network (GFCC-MDNN).

As depicted in Figure 12, the recognition accuracy of the proposed modified DNN is higher than that of the GMM. Although the GMM can recognize underwater acoustic targets, it is a shallow recognition model in which the recognition accuracy is relatively low. Meanwhile, the recognition accuracy of the GFCC-MDNN was higher than that of the MFCC-MDNN, which proves that the GFCC has strong adaptability to modify the DNN in this paper, and it is suitable for underwater acoustic target recognition compared to MFCC.

To further demonstrate the recognition accuracy of the proposed the MFF-MDNN in this paper, Figure 13 shows the recognition accuracy of the proposed modified the DNN recognition method using GFCC feature extraction (GFCC-MDNN) [18], the proposed modified DNN recognition method using MFCC and MEMD feature extraction (MM-MDNN) [16], and the proposed MFF-MDNN.

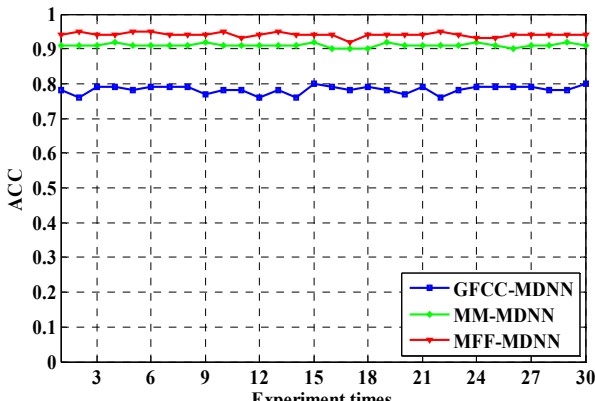

**Figure 13.** The recognition accuracy of the GFCC-MDNN, the MM-MDNN, and the multi-dimensional fusion features and a modified deep neural network MFF-MDNN.

As seen in Figure 13, the recognition accuracy of the proposed MFF-MDNN was higher than the other method. Therefore, the proposed MFF-MDNN has better recognition accuracy for underwater acoustic target recognition in this paper.

To describe more clearly the reliability and stability of the proposed MFF-MDNN method, Table 2 shows recognition results of 30 experiments with a different training set and testing set in the dataset when the maximum iteration was 800, which includes the MFCC-GMM, GFCC-GMM, MFCC-MDNN, GFCC-MDNN, MM-MDNN, and the proposed MFF-MDNN.

**Table 2.** The recognition results of 30 experiments with a different training set and testing set in the dataset.

| Experiment Times | MFCC-GMM | GFCC-GMM | MFCC-MDNN | GFCC-MDNN | MM-MDNN | MFF-MDNN |
|---|---|---|---|---|---|---|
| 1 | 0.59 | 0.61 | 0.74 | 0.78 | 0.91 | 0.94 |
| 2 | 0.60 | 0.62 | 0.72 | 0.76 | 0.91 | 0.95 |
| 3 | 0.62 | 0.65 | 0.76 | 0.79 | 0.91 | 0.94 |
| 4 | 0.64 | 0.60 | 0.74 | 0.79 | 0.92 | 0.94 |
| 5 | 0.58 | 0.64 | 0.73 | 0.78 | 0.91 | 0.95 |
| 6 | 0.58 | 0.62 | 0.73 | 0.79 | 0.91 | 0.95 |
| 7 | 0.60 | 0.65 | 0.72 | 0.79 | 0.91 | 0.94 |
| 8 | 0.61 | 0.64 | 0.71 | 0.79 | 0.91 | 0.94 |
| 9 | 0.58 | 0.61 | 0.72 | 0.77 | 0.92 | 0.94 |
| 10 | 0.58 | 0.63 | 0.74 | 0.78 | 0.91 | 0.95 |
| 11 | 0.60 | 0.64 | 0.74 | 0.78 | 0.91 | 0.93 |
| 12 | 0.58 | 0.65 | 0.72 | 0.76 | 0.91 | 0.94 |
| 13 | 0.58 | 0.65 | 0.75 | 0.78 | 0.91 | 0.95 |
| 14 | 0.58 | 0.59 | 0.74 | 0.76 | 0.91 | 0.94 |
| 15 | 0.59 | 0.60 | 0.72 | 0.80 | 0.92 | 0.94 |
| 16 | 0.58 | 0.62 | 0.73 | 0.79 | 0.90 | 0.94 |
| 17 | 0.60 | 0.61 | 0.74 | 0.78 | 0.90 | 0.92 |
| 18 | 0.58 | 0.62 | 0.72 | 0.79 | 0.90 | 0.94 |
| 19 | 0.57 | 0.61 | 0.75 | 0.78 | 0.92 | 0.94 |
| 20 | 0.59 | 0.60 | 0.74 | 0.77 | 0.91 | 0.94 |
| 21 | 0.58 | 0.63 | 0.72 | 0.79 | 0.91 | 0.94 |
| 22 | 0.58 | 0.61 | 0.71 | 0.76 | 0.91 | 0.95 |
| 23 | 0.58 | 0.60 | 0.73 | 0.78 | 0.91 | 0.94 |
| 24 | 0.60 | 0.62 | 0.74 | 0.79 | 0.92 | 0.93 |
| 25 | 0.61 | 0.63 | 0.72 | 0.79 | 0.91 | 0.93 |
| 26 | 0.58 | 0.61 | 0.74 | 0.79 | 0.90 | 0.94 |
| 27 | 0.58 | 0.60 | 0.70 | 0.79 | 0.91 | 0.94 |
| 28 | 0.60 | 0.62 | 0.72 | 0.78 | 0.91 | 0.94 |
| 29 | 0.57 | 0.64 | 0.74 | 0.78 | 0.92 | 0.94 |
| 30 | 0.59 | 0.64 | 0.72 | 0.80 | 0.91 | 0.94 |

As seen from Table 2, the recognition accuracy of the proposed MFF-MDNN method was higher than that of the other algorithms in the 30 experiments with a different training set and testing set in the dataset when the maximum iteration was 800. In this paper, the proposed multi-dimensional fusion features method can describe underwater acoustic signals from multiple angles. It combines the advantages of the GFCC with MEMD, which is more suitable for underwater acoustic target recognition than the single feature. Moreover, GMM was used to extract the statistical parameters of the feature matrix, which modify the structure of DNN. It can reduce redundant features and further improve recognition accuracy. From the above comparative experiments in Tables 1 and 2, the following results can be drawn. The proposed MFF-MDNN method can improve accuracy when the dataset has underwater background noise with weak targets. Therefore, the recognition results demonstrate that the proposed MFF-MDNN method has higher accuracy and strong adaptability over other methods.

## 5. Conclusions

In this paper, a combination of multi-dimensional fusion features and modified DNN method was proposed to recognize underwater acoustic targets. The problem where the single feature could not describe underwater acoustic signals well was solved effectively, using the GFCC and MEMD to extract multi-dimensional features. On this basis, a dimension reduction method was proposed, which fused the multi-dimensional features in the same time dimension. It could obtain multi-dimensional fusion features in the original underwater acoustic signals. In addition, the problem of many redundant features in the processing of recognition was solved by utilizing GMM to modify the structure of DNN. It could improve recognition accuracy. The proposed underwater acoustic target recognition method was employed on a dataset, which was divided into six categories, including four types of

ships, underwater mammals, and underwater background noise with weak targets. The average recognition accuracies of the proposed MFF-MDNN, MM-MDNN, GFCC-MDNN, MFCC-MDNN, GFCC-GMM, and MFCC-GMM were 94.3%, 91.1%, 78.2%, 73.1%, 62.2%, and 59.0%, respectively, when the maximum iterations were 800. The recognition results showed that the method had good validity and adaptability. However, MFF-MDNN has a higher time complexity due to the algorithm principle of MEMD and the particularity of underwater acoustic signals. Further studies can be conducted in the future in this direction.

**Author Contributions:** Conceptualization, X.W., A.L., Y.Z., and F.X.; Funding acquisition, X.W.; Investigation, Y.Z. and F.X.; Methodology, A.L.; Project administration, X.W.; Software, Y.Z. and F.X.; Supervision, X.W. and F.X.; Writing—original draft, A.L. and Y.Z.; Writing—review and editing, X.W., A.L., Y.Z., and F.X.

**Funding:** This work was supported by the National Natural Science Foundation of China grant number 41876110 and Fundamental Research Funds for the Central Universities grant number 3072019CFT0602.

**Acknowledgments:** The authors are grateful to the guest editors and anonymous reviewers for their constructive comments based on which the presentation of this paper has been greatly improved.

**Conflicts of Interest:** The authors declare no conflict of interest.

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
