# Peer review of "Underwater Acoustic Target Recognition: A Combination of Multi-Dimensional Fusion Features and Modified Deep Neural Network"

_remotesensing, doi:10.3390/rs11161888_

Round 1

Reviewer 1 Report

This paper presents a signal-processing method, based on the combination of Multi-Dimensional Fusion Features (MFF) and Modified Deep Neural Network (MDNN), for the recognition of underwater acoustic target. The detailed implementation of the method is described. The effectiveness of the method is demonstrated by analyzing six different types of experimental datasets.  

The paper presents a framework of the modified signal processing method for an underwater acoustic target recognition, which is suitable for publication in the journal of Remote Sensing. Nevertheless, the manuscript can be largely improved in the following areas: 

There are numerous English grammar errors. Many places are not readable. The text should be carefully re-edited. 

The limitations of the present method (developed in this paper) should be discussed. Are there any conditions under which the effectiveness of the method would reduced? 

The definition of the variable b in equation (2), differs from its definition in equation (4).  Is the variable b in equation (2) the same as that in equation (4) or not?

i and l in equation (6) are not specified or defined.

On the line 148, "... is the amplitude of H(t)", ... This is incorrect. It should be "the phase of H(t)"

Author Response

Response to Reviewer 1 Comments

Point 1: There are numerous English grammar errors. Many places are not readable. The text should be carefully re-edited. (Reviewer 1)

Response: According to comment of reviewer 1 and deep thinking, the authors carefully checked the English grammar errors and corrected them. In addition, Dr. Yin Dan, who is Visiting Scholar from school of Computer Science & Engineering, The University of New South Wales, also help us to revise the present manuscript. Please see revised manuscript.

Point 2: The limitations of the present method (developed in this paper) should be discussed. Are there any conditions under which the effectiveness of the method would reduce?  (Reviewer 1)

Response: According to comment of reviewer 1 and deep thinking, to make the description more clearly reflect the work, authors add “However, MFF-MDNN has a higher time complexity due to the algorithm principle of MEMD and the particularity of underwater acoustic signals. It can be conducted in this direction in the future.” In the conclusion to discuss the limitations of the present method. Please see revised manuscript.

Point 3:  The definition of the variable b in equation (2), differs from its definition in equation (4).  Is the variable b in equation (2) the same as that in equation (4) or not? (Reviewer 1)

Response: According to comment of reviewer 1 and deep thinking, “is the attenuation factor of the filter,” in section 2.1 and equation (2) makes description unclear. They are revised as is the attenuation factor of the filter,” and in the third paragraph of section 2.1. This makes the description more clearly reflect the work. Please see revised manuscript.

Point 4:  i and l in equation (6) are not specified or defined. (Reviewer 1)

Response: According to comment of reviewer 1 and deep thinking, equation (6) “” is revised as “” where i is the same as defined in equation (5). This makes the description more clearly reflect the work. Please see revised manuscript. 

Point 5: On the line 148, "... is the amplitude of H(t)", ... This is incorrect. It should be "the phase of H(t)" (Reviewer 1)

Response: According to suggestion of reviewer 1 and literature review, “... is the amplitude of H(t)” is revised as “the phase of H(t)” on the line 147. Please see revised manuscript.

Reviewer 2 Report

this paper considers an Underwater Acoustic Target Recognition scheme by combining Multi-dimensional Fusion Features and a Modified Deep Neural Network. This work is of interest and is within the journal's scope.

The concept and the performance attained are promising

Author Response

Response to Reviewer 2 Comments

Point 1: his paper considers an Underwater Acoustic Target Recognition scheme by combining Multi-dimensional Fusion Features and a Modified Deep Neural Network. This work is of interest and is within the journal's scope.

The concept and the performance attained are promising (Reviewer 2)

Response: According to comment of reviewer 2, thank you for your affirmation, the authors have also improved other deficiencies regarding the paper. Please see revised manuscript.

Reviewer 3 Report

The paper presents an underwater target recognition approach that is referred as MFF-MDNN in this paper. Using experimental results the approach is shown to be superior to the state of teh art. 

The authors did a great job in presenting a detailed literature review in clear terms. The rest of the paper is however sloppily written. Although the paper is neatly structured, it is very verbose at times while missing key details. Also, the paper needs many more proof readings to get rid of unclear language in the paper: some examples are provided at the end of the review.

In Section 2.3.2, figure 5 and algorithm 1, the MFF part is explained. The authors mention dimensionality reduction but they neither provide details on the underlying algorithm used nor the absolute numbers such as the compression factor of the dimensionality reduction algorithm. On the other hand GFCC and MEMD which could have been referred from other papers were derived in the paper. Similar comments for section 3 describing the MDNN part. GMM and MLE, EM were explained and derived which could have been found in any standard machine learning textbook. Without this verbosity the paper would have been a lot shorter and extremely reader friendly. 

Another issue in section 3 is the lack of clear explanation about why GMM was necessary before feeding data into a DNN. DNNs these days are shown to outperform classical machine learning techniques which also included GMMs. Therefore a slightly larger DNN with a couple more layers might also learn what a GMM model would learn. In this light, more details arguing for the necessity of a GMM+DNN could have improved the paper. 

Experimental results sections has been well written and explains the difference between MFCC and GFCC, EMD and MDMD, and also shows the accuracy of MFF-MDNN relative to a number of other variants of the algorithm. I especially liked that way, the difference between MFCC and GFCC has been presented. If this explanations is added to section 2, it will be incredibly helpful for the readers in understanding the difference early on while reading the paper. 

One concern about Table 2 in experimental results is that the machine learning part of the algorithm is separately applied to train and test each dataset independently. Why is that ? The regular practice is that either by combining all the datasets or by just training on one of the datasets, a single model is learned. To test the performance of the model, it is tested on data from all datasets. The approach presented in the paper could have been right if the set of output targets is different for each dataset but that is not the case here (or it is not mentioned in the paper). Instead when one trains a separate model on each dataset and evaluates test accuracy on the same dataset, there is very goo chance of overfitting the model to that particular dataset and hence the accuracy numbers obtained may not be reliable. 

Overall, the paper is clearly organised but verbose and missing important information at times. I hope the authors will address the issues in the final version if the paper is accepted for publication. The paper also needs extensive proof reading and grammatical checks and sentence clarity; I have listed a number of examples below.

Line 54 - 55: Sentence not grammatical. "it cannot accurate description original..." should have been "it cannot accurately describe original..."

Line 64-65: I did not understand what this sentence means. 

Line 93-94: Sentence could be grammatically better

Line 97: "Underwater mammals" is preferred to  "underwater mammal animals"

Line 186-187: Sentence could be improved - "many redundant features are commonly existed"

Line 258 - "Gaussian" instead of "Gauss"

Line 284 - 285 - "may be" , after "distinct" ?

Author Response

Response to Reviewer 3 Comments

Point 1: In Section 2.3.2, figure 5 and algorithm 1, the MFF part is explained. The authors mention dimensionality reduction but they neither provide details on the underlying algorithm used nor the absolute numbers such as the compression factor of the dimensionality reduction algorithm. On the other hand, GFCC and MEMD which could have been referred from other papers were derived in the paper. Similar comments for section 3 describing the MDNN part. GMM and MLE, EM were explained and derived which could have been found in any standard machine learning textbook. Without this verbosity, the paper would have been a lot shorter and extremely reader friendly. (Reviewer 3)

Response: According to comment of reviewer 3 and deep thinking, the specific revises are as follows:

The authors addThe frame length equals the ratio of one-dimensional feature (IE or IF) to GFCC feature length.” in the section 2.3.2 which explains specifically how to set the parameters. To make the paper a lot shorter and extremely reader friendly, the authors delete part of the explanation about GMM and EM in section 3.

Please see revised manuscript.

Point 2: Another issue in section 3 is the lack of clear explanation about why GMM was necessary before feeding data into a DNN. DNNs these days are shown to outperform classical machine learning techniques which also included GMMs. Therefore, a slightly larger DNN with a couple more layers might also learn what a GMM model would learn. In this light, more details arguing for the necessity of a GMM DNN could have improved the paper. (Reviewer 3)

Response: According to comment of reviewer 3 and deep thinking, to make the description more clearly reflect the work, authors give more explains about the MDNN in the section 3 first paragraph When DNN is directly used to recognize underwater acoustic target, due to the direct input of the features of the underwater acoustic signals, there indeed exist many redundant features in processing of recognition. GMM can extract the statistical parameters of the underwater acoustic signals, which can reduce the length of the underwater acoustic signals. GMM can reduce the size of the DNN model, can reduce redundant features and further improve recognition accuracy. Therefore, GMM is used to modify the structure of DNN in this paper. The modified DNN which is shown in Figure6, it is a fully connected feedforward neural network with multiple hidden layers, and it can better accomplish underwater acoustic target recognition tasks. Please see revised manuscript.

Point 3: Experimental results sections has been well written and explains the difference between MFCC and GFCC, EMD and MDMD, and also shows the accuracy of MFF-MDNN relative to a number of other variants of the algorithm. I especially liked that way, the difference between MFCC and GFCC has been presented. If this explanations is added to section 2, it will be incredibly helpful for the readers in understanding the difference early on while reading the paper. (Reviewer 3)

Response: According to comment of reviewer 3 and deep thinking, to make helpful for the readers in understanding the difference early, the authors add these explanations to section 2. Please see revised manuscript.

Point 4: One concern about Table 2 in experimental results is that the machine learning part of the algorithm is separately applied to train and test each dataset independently. Why is that ? The regular practice is that either by combining all the datasets or by just training on one of the datasets, a single model is learned. To test the performance of the model, it is tested on data from all datasets. The approach presented in the paper could have been right if the set of output targets is different for each dataset but that is not the case here (or it is not mentioned in the paper). Instead when one trains a separate model on each dataset and evaluates test accuracy on the same dataset, there is very goo chance of overfitting the model to that particular dataset and hence the accuracy numbers obtained may not be reliable. (Reviewer 3)

Response: According to comment of reviewer 3and deep thinking, to make the description more clearly reflect the work, authors conducted 30 experiments, each time randomly dividing the data set into two parts, One part is used as a training set and the other part is used as a test set with a ratio of 7 to 3. It should conform to the second case described in reviewer 3.

Point 5: Line 54 - 55: Sentence not grammatical. "it cannot accurate description original..." should have been "it cannot accurately describe original..." (Reviewer 3)

Response: According to suggestion of reviewer 1 and literature review, “it cannot accurate description original...” is revised as “it cannot accurately describe original...” on the line 54-55. Please see revised manuscript.

Point 6: Line 64-65: I did not understand what this sentence means. (Reviewer 3)

Response: According to suggestion of reviewer 1 and literature review, “Gaussian mixture model (GMM) can obtain superior results.” Is revised as “Gaussian mixture model (GMM) is widely used in the field of recognition and has achieved good results.” On the line 64-65. Please see revised manuscript.

Point 7: Line 93-94: Sentence could be grammatically better. (Reviewer 3)

Response: According to suggestion of reviewer 1 and literature review, “GMM is used to modify the structure of DNN to improve recognition accuracy.” Is revised as “GMM is used to modify the structure of DNN to improve the accuracy of recognition.” On the line 93-94. Please see revised manuscript.

Point 8: Line 97: "Underwater mammals" is preferred to  "underwater mammal animals". (Reviewer 3)

Response: According to suggestion of reviewer 1 and literature review, “underwater mammal animals” is revised as “underwater mammals” on the line 97. Please see revised manuscript.

Point 9: Line 186-187: Sentence could be improved - "many redundant features are commonly existed" (Reviewer 3)

Response: According to suggestion of reviewer 1 and literature review, “many redundant features are commonly existed” is revised as “there indeed exist many redundant features” on the line186-187. Please see revised manuscript.

Point 10: Line 258 - "Gaussian" instead of "Gauss" (Reviewer 3)

Response: According to suggestion of reviewer 3, “Gauss white noise is added into Figure 7” is revised as “Gaussian white noise is added into Figure 7” on line 261. Please see the revised manuscript.

Point 11: Line 284 - 285 - "may be" , after "distinct" ? (Reviewer 3)

Response: According to suggestion of reviewer 3, “While the MEMD can decompose the original underwater acoustic signals more distinct.” is revised as “While the MEMD can decompose the original underwater acoustic signals more distinct may be.” on line 287. Please see the revised manuscript.

Round 2

Reviewer 3 Report

The authors did a good job in addressing all the requested changes and should be commended for their research.